# Modelling Drought Risk Using Bivariate Spatial Extremes: Application to the Limpopo Lowveld Region of South Africa

**Murendeni Maurel Nemukula** [1,2,3,†] , **Caston Sigauke** [1,*,†] , **Hector Chikoore** [3] **and Alphonce Bere** [1]

1   Department of Mathematical and Computational Sciences, University of Venda, Private Bag X5050, Thohoyandou 0950, South Africa
2   School of Mathematics, Statistics and Computer Science, University of KwaZulu-Natal, Private Bag X54001, Durban 4000, South Africa
3   Department of Geography and Environmental Studies, University of Limpopo, Private Bag X1106, Sovenga 0727, South Africa
*   Correspondence: caston.sigauke@univen.ac.za; Tel.: +27-15-962-8135
†   These authors contributed equally to this work.

**Abstract:** Weather and climate extremes such as heat waves, droughts and floods are projected to become more frequent and intense in several regions. There is compelling evidence indicating that changes in climate and its extremes over time influence the living conditions of society and the surrounding environment across the globe. This study applies max-stable models to capture the spatio–temporal extremes with dependence. The objective was to analyse the risk of drought caused by extremely high temperatures and deficient rainfall. Hopkin's statistic was used to assess the clustering tendency before using the agglomerative method of hierarchical clustering to cluster the study area into $n = 3$ temperature clusters and $n = 3$ precipitation clusters. For the precipitation and temperature data, the values of Hopkin's statistic were 0.7317 and 0.8446, respectively, which shows that both are significantly clusterable. Various max-stable process models were then fitted to each cluster of each variable, and the Schlather model with several covariance functions was found to be a good fit on both datasets compared to the Smith model with the Gaussian covariance function. The modelling approach presented in this paper could be useful to hydrologists, meteorologists and climatologists, including decision-makers in the agricultural sector, in enhancing their understanding of the behaviour of drought caused by extremely high temperatures and low rainfall. The modelling of these compound extremes could also assist in assessing the impact of climate change. It can be seen from this study that the size, including the topography of the location (cluster/region), provides important information about the strength of the extremal dependence.

**Keywords:** clustering; climate extremes; extremal dependence; spatial dependence

## 1. Introduction

### 1.1. Overview

Drought is a manifestation of climate variability that is ranked near the top of all hydro-meteorological hazards in terms of the number of people it affects globally (Lyon [1]). However, drought occurs in most climates (except for semipermanent arid regions where it does not have meaning) and can lead to significant socio-economic and environmental impacts (Nembilwi et al. [2]). The extremely high temperatures that simultaneously occur with drought contribute to crop losses and widespread livestock mortality based on the stresses on regional water supplies, as emphasised in Lyon [1]. Throughout this paper, temperature represents air temperature.

### 1.2. Literature Review

The Limpopo Province, located in the northeast of South Africa, is largely semiarid, making it frequently vulnerable and affected by meteorological drought (Mathivha et al. [3]).

Nembilwi et al. [2] describe drought as a normal but complex, slow-onset and recurring aspect of global climate, particularly in the semiarid subtropics. More recently, the occurrence of the 2015/16 El Niño drought season was the hottest recorded drought period and one of the driest, accounting for significant repercussions to livelihoods and economic development (Baudoin et al. [4]). Drought events lead to devastating impacts that affect water levels, crop yields, livestock herds and rural livelihoods. It is therefore important to guard against the risk of the occurrence, nature and impacts of drought hazards, including the vulnerability of rural communities in Limpopo Province. It is, therefore, important to develop adaptation strategies that can be used to cope with natural hazards caused by drought. This leads to a need for thorough and accurate statistical models capable of modelling the asymptotic behaviour of thin- or heavy-tailed distributions of compound meteorological extreme events.

In modelling extremal events, the traditional risk assessment methods are typically based on one driver or hazard at a time, potentially leading to the underestimation of risk (Zscheischler et al. [5]). This is because processes that cause extreme events often interact and are spatially or temporally dependent. Zscheischler et al. [5] further emphasise that the combination of processes such as climate drivers and hazards leading to a significant impact can be considered a compound event, which is formally defined by Hao et al. [6] as a simultaneous or sequential occurrence of multiple extremes at single or multiple locations that may exert even more significant impacts on society or the environment.

In their paper, Lellyett et al. [7] proposed a research direction which can improve the early warning of drought. The authors argued that the use of index- and impact-based forecasting models should consider seasonal, including multiyear time scales. In support of the use of drought indices, Ndayiragije and Li [8], in their study, focussed on drought management mitigation measures, including management of drought risk in assessing the effectiveness of drought indices.

Using South African data from a winter rainfall zone in the Western Cape Province, Conradie et al. [9] assessed the spatio-temporal patterns of drought intensity. Their results suggested significant variability in the spatial extent of the drought. It is now generally agreed in the literature that droughts as a result of climate change globally are going to be more frequent and severe (Ferreira et al. [10]). Using two sites in the Limpopo Province of South Africa, Ferreira et al. [10] assessed the spatio-temporal variability of drought patterns including their impacts in maize production.

Compound extreme events such as low precipitation and high-temperature result in high drought risks. Understanding the joint distribution of such extreme events helps decision-makers quantify the magnitude of their collective impact. Using copula models, Esit and Yuce [11] constructed spatial distributions of drought risk–return periods under four scenarios of drought risk, which are light, moderate, severe and extreme drought. The authors argued that drought should be assessed based on several variables. Using large ensemble simulations, Singh et al. [12] assessed individual and joint variations of extreme precipitation and temperature. Canadian data were used in the study, with results showing increases in extremely hot temperatures in central and southeastern Canada. In the western coastal regions of the country, results suggested increases in wet extremes.

The focus of this paper was to study the interdependence of annual average precipitation and maximum temperature in the Lowveld region of Limpopo Province in South Africa. This was followed by an analysis of the extent to which the simultaneous occurrence of these extremal events jointly contributes towards the risk of the occurrence of drought in the area of the study. The study was motivated by the changing climate, its current rates, frequency, duration and intensity and its life-threatening impacts, which are undoubtedly abnormal and globally worrisome. This study demonstrates the applicability of the agglomerative method of hierarchical clustering to cluster the study area before fitting max-stable process models.

Many meteorological processes to which the extreme values can be damaging are inherently spatial (Wadsworth and Tawn [13]). For this reason, much recent interest has been

in the statistical modelling of spatial extremes. In line with this, the rainfall distribution in the Lowveld region of Limpopo Province is characterised by high spatial and temporal variability, which may be partially attributed to strong spatial gradients in elevation in the area (Nembilwi et al. [2]). However, there are two types of concerns when dealing with extreme values of spatial processes (Davison et al. [14]). These are the accurate inferences for site-wise marginal distributions and assessing the spatial dependence of the extreme values.

The first issue is usually addressed because classical extreme value theory relies on max-stability (Ribatet [15]). Approaches of this kind typically focus on the spatial smoothness of marginal distribution parameters but do not model spatial dependence. However, according to Ribatet [15], certain fundamental challenges arise due to the restrictive assumptions that must be made when using max-stable processes to model dependence for spatial extremes. For instance, it must be assumed that the dependence structure of the observed extremes is compatible with a limiting model that holds for all events that are more extreme than those that have already occurred (Davison et al. [14]). This problem has long been acknowledged in the context of finite-dimensional bivariate extremes, particularly when data display dependence at detectable levels but are independent in the limit (Davison et al. [14]). A flexible class of models suitable for such data in a spatial context has been proposed in Wadsworth and Tawn [13].

In connection with modelling meteorological drought patterns using compound extremes, it is emphasised by Zscheischler et al. [5] that extremely high temperatures and insufficient rainfall, viewable as compound extremes, are among the factors determining drought intensity. For instance, Chikoore [16] argues that the expected global annual rainfall is roughly 860 mm, which is way above South Africa's average rainfall of 450 mm. Due to this limitation, any disturbance in rainfall patterns can profoundly impact the livelihoods and environment within the Lowveld region of Limpopo Province. Furthermore, Chikoore and Jury [17] argue that water deficits arise from the imbalance of seasonal rainfall and constantly high evaporation.

To this effect, as the demand for water grows, resources tend to be stressed to scarcity. Bivariate spatial compound extremes models are considered a relevant framework for drought based on the simultaneous occurrence of extremely high temperatures and low rainfall rates. This study used models for spatial extremes in modelling compound extremes when extremal dependence structure may vary with distance. Spatial extremes are considered suitable approaches for quantifying spatial dependence in various bivariate processes, including compound extremes.

The rest of the paper is organised as follows: Section 2 describes the regional setting, the study area and the variables, including the data sources. The methodology is discussed in Section 3, whereas the empirical results are presented in Section 4. The discussion of the results is presented in Section 5, and Section 6 offers conclusions.

## 2. Study Area

The Limpopo Province in South Africa is located in the northeast of South Africa, bordering Botswana in the west, Zimbabwe in the north and Mozambique in the east (Figure 1). The Limpopo Province in South Africa has three distinct climate regions: the Lowveld, characterised by a semiarid climate; the middle and Highveld, considered semiarid with an escarpment that experiences subhumid climate (Usman and Reason [18]). The chosen study area for this paper was the Lowveld region between latitude $-22°$ and $-24°$ and longitude $30°$ and $33°$. This area is a subtropical and semiarid region with warm temperatures all year round, receiving an average of 500 mm annual rainfall during the summer season between October and March (Chikoore and Jury [17]). This study area experiences high climate variability from season to season and is vulnerable to extreme weather conditions, such as high temperatures and dry conditions that lead to drought.

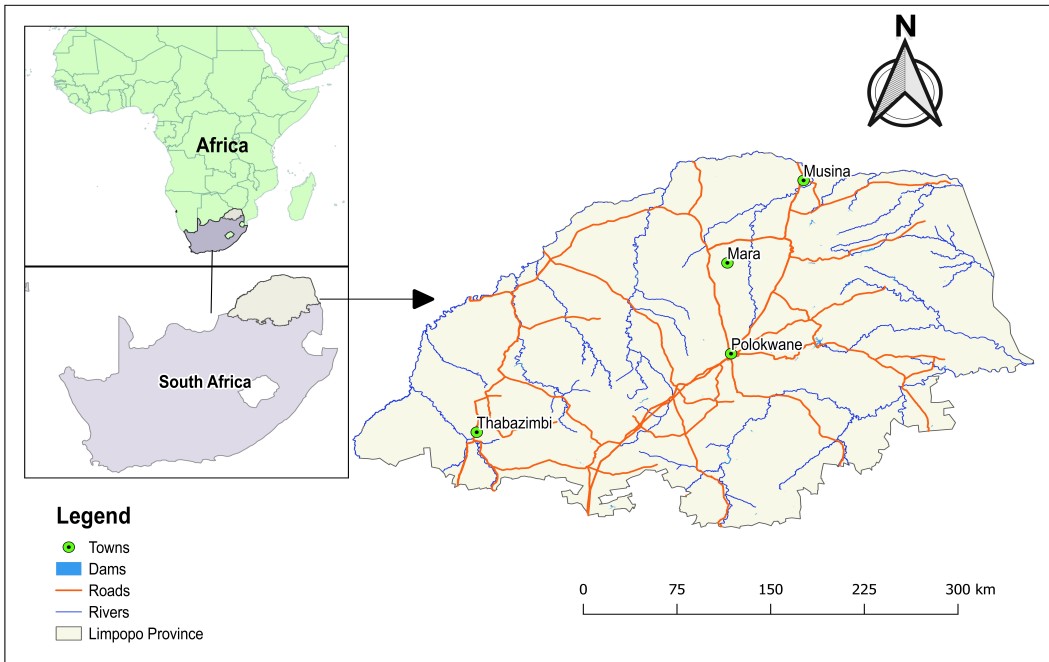

**Figure 1.** Map of Africa, South Africa and Limpopo Province with the main towns. Source: Authors' creation.

On most days, the selected study area experiences long sunny days and dry weather conditions (Chikoore [16]). The study area experiences very hot conditions in summer (October to March), with average temperatures rising to 27 °C and falling to about 20 °C in winter. Most of the precipitation occurs during the austral summer, and annual rainfall ranges from about 400 to 600 mm over most of the province (Chikoore [16]). The communities in the Limpopo region may need to develop resilience and adapt to long-term climate changes, such as increased seasonal temperature and altered patterns of precipitation. However, they tend to be heavily stressed by the frequency of these extreme weather events (Usman and Reason, [18]).

The maps that geographically summarise the average annual precipitation and the average maximum temperature within the study area are given in Figures 2 and 3, respectively. These maps further show clusters for annual average precipitation and maximum temperature within the study area. Visual inspection of the study area in Figure 2 appears to have three subregions. Between 29.4° and 30°, the precipitation ranges from about 351.5 to 401.5 mm. For longitudes 30° to 31°, the precipitation ranges from 301.5 to 351.5 mm, while for longitudes 31° to 32.4°, the precipitation ranges from 201.5 to 301.5 mm, similar to the temperature in the study area given in Figure 3. For longitudes 29.4° to 30° the temperature ranges from 30.2 °C to 33.3 °C; for longitudes 30° to 31° the temperature is between 33.3 °C and 34.8 °C and ranges between 34.8 °C and 36.3 °C for longitudes 31° to 32.4°.

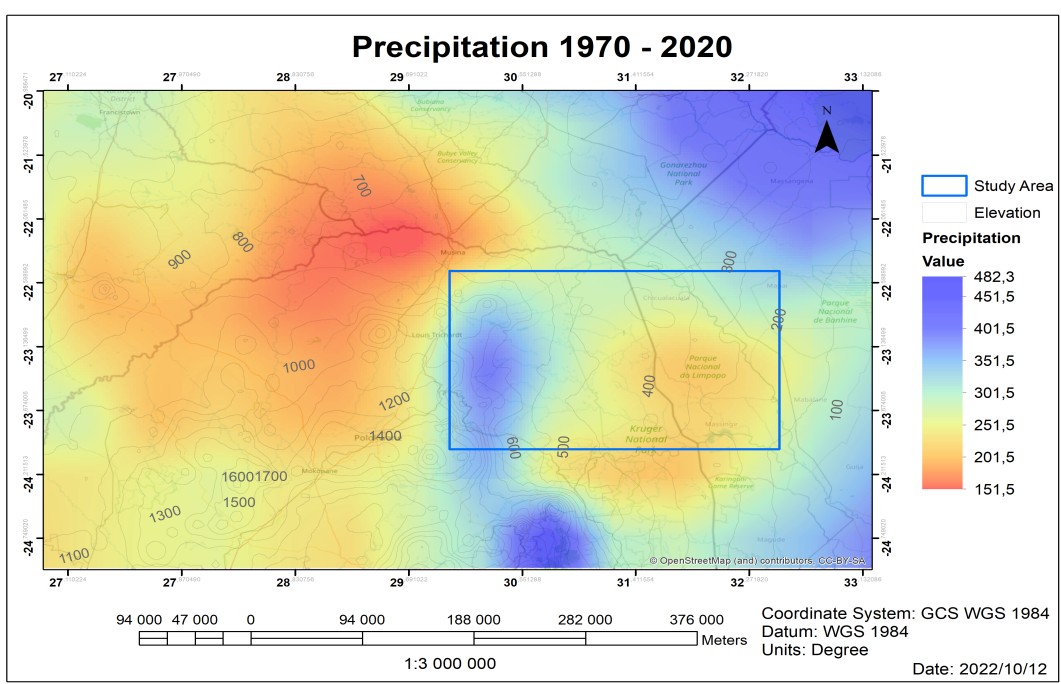

**Figure 2.** Map showing the annual average precipitation of the study area in the Lowveld region of Limpopo. Source: Authors' creation.

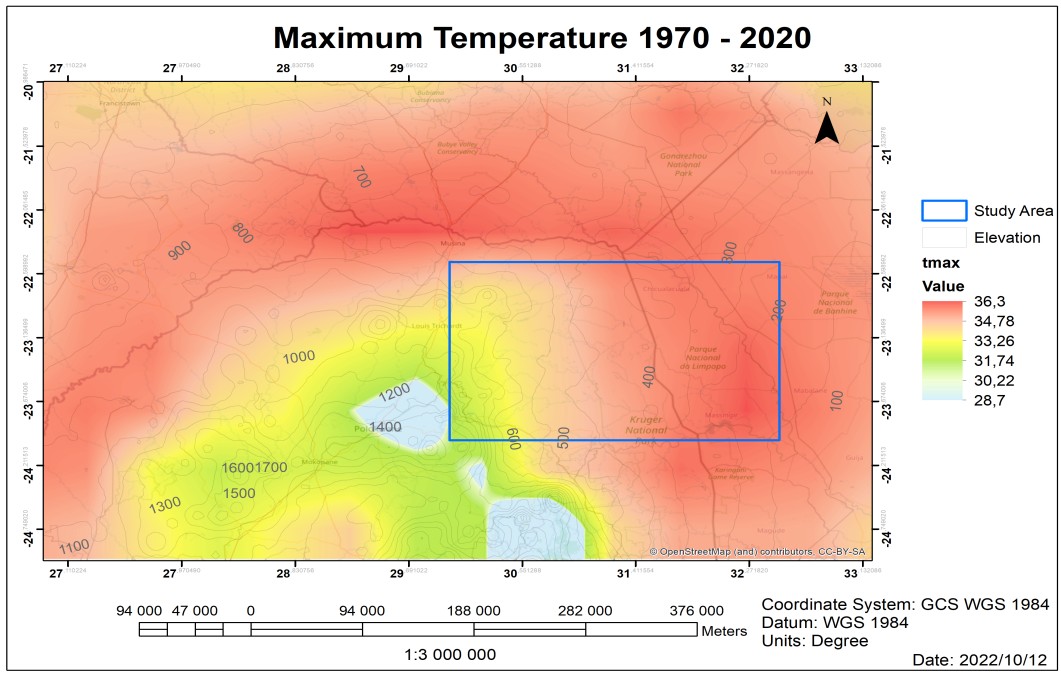

**Figure 3.** Map showing the annual maximum temperature of the study area in the Lowveld region of Limpopo. Source: Authors' creation.

*Description of Variables and Data Sources*

The study estimates the extremal dependence of annual average precipitation and maximum temperature. This helps quantify the joint impact of these two weather variables on the risk of drought occurrence. The data comprise the Climatic Research Unit gridded Time Series (CRU TS), a widely used climate dataset in climate research. The CRU data are available on a 0.5° latitude by 0.5° longitude grid over all land domains of the world, except Antarctica (Harris et al. [19]). The data were collected from the KNMI climate explorer web portal URL (accessed on 31 October 2022) https://climexp.knmi.nl/start.cgi.

In this paper, the gridded data are preferred over the station data for the resolution wherein the effect of the presence of mountains and plantations on the climate can be visualised and understood. We then selected 24 grid points from which the monthly average precipitation and maximum temperatures were recorded at 0.5° resolution. These were collected from 1 January 1970 to 31 December 2020, each year for 50 years. These observations were recorded for monthly average precipitation and maximum temperature for all 24 grid points. The analyses in this study were performed using the R programming language (R Core Team, 2013). The max-stable process models and most spatial extremes objects were computed based on the 'SpatialExtremes' R package developed by Ribatet [20].

## 3. Methodology

The flowchart of the structure of the modelling process is shown in Figure 4. The data consist of temperature and precipitation. The preprocessing stage consists of cleaning the data, imputing missing values and detecting outliers.

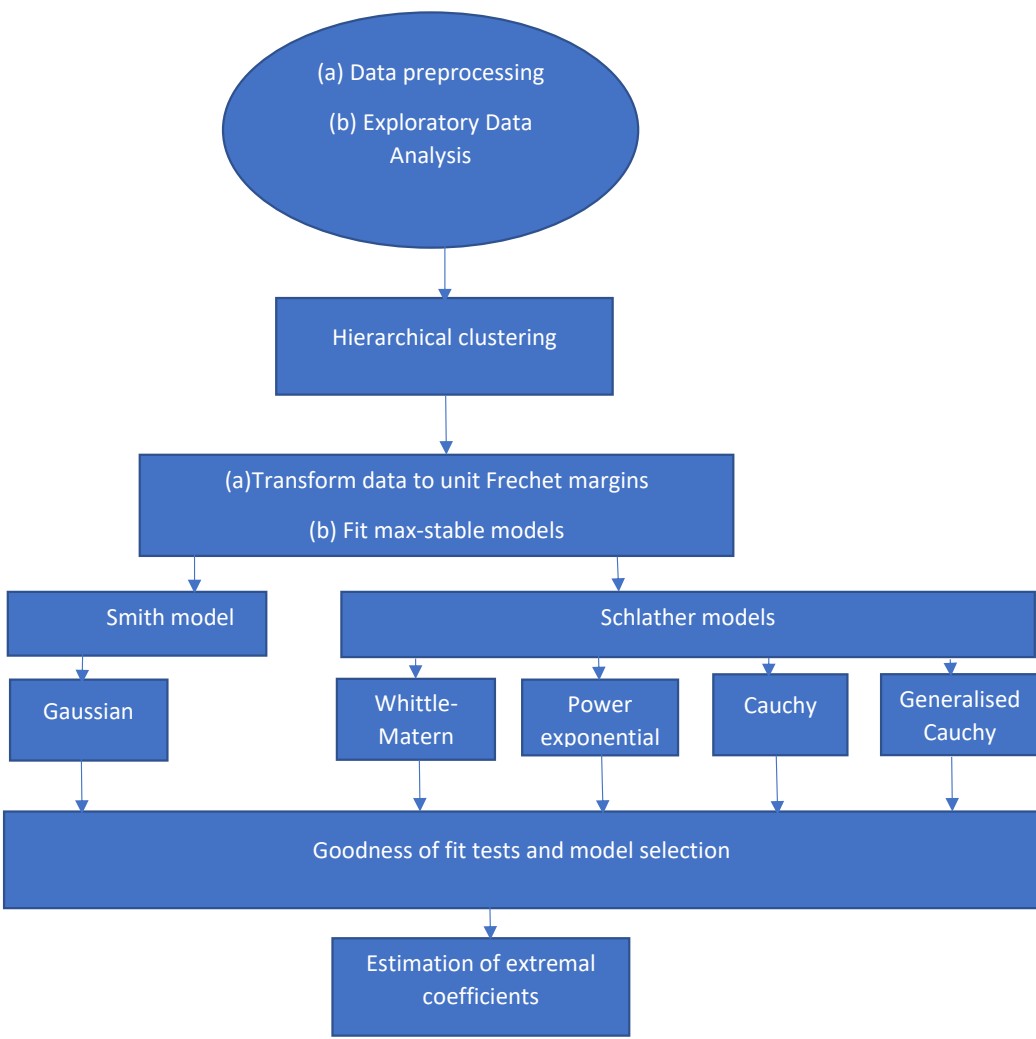

**Figure 4.** Flowchart showing the modelling process. Source: Authors' creation.

### 3.1. Hierarchical Clustering

Hierarchical clustering is performed based on the F-madogram space in various applications. However, the regions must be defined in a euclidean space to maintain high accuracy instead of the F-madogram. This problem is addressed by including an additional classification step which is also important for classifying locations without stations (Saunders et al. [21]). The classification procedure is important in both hierarchical

and nonhierarchical clustering. The classification step is also important for identifying boundaries between two clusters during the predictions.

Saunders et al. [21] prefer using a weighted K-nearest neighbour classifier to classify grid points. In the standard k-nearest neighbour classification, the points are classified without using weights. A nonlinear relationship between the F-madogram and euclidean distance makes a weighted classifier more suitable for various applications (Nielsen [22]). However, many authors in the literature prefer using an inverse-weighted kernel. During the selection of the number of nearest neighbours, the limitation usually encountered is the variance bias trade-off, which can be addressed by choosing clusters well separated in Euclidean space.

The hierarchical clustering approach is characterised by creating an ordered sequence called a hierarchy of partitions. To assess the dependence structure of temperature and precipitation in this paper, it is important to consider the use of suitable clustering methods before fitting the max-stable models. A hierarchical clustering approach that is believed to produce a hierarchy of partitions with an honest insight into our application gains preference. It can be interpreted as the partitions of points based on strong dependence to weaker dependence.

This ordered sequence of partitions is represented graphically using a dendrogram showing the hierarchical relationship between objects (Saunders et al. [21]). Each point is considered a cluster or leaf in the dendrogram, whereas the branches in the dendrogram are formed by successfully combining leaves and other branches until all points are grouped. A new partition of the points is then induced for each merge. Therefore, the successful merging of branches creates the ordered partition of points, as Saunders et al. [21] emphasised. To decide the number of branches to be merged, the definition of distance needs to be extended from between two points to include the distance between two groups of points. Let $C_k$ and $C_{k'}$ be two different clusters of points with $x$ representing the points in each of the clusters. According to Saunders et al. [21], the average linkage criterion

$$d(C_k, C_{k'}) = \frac{1}{|C_k||C_{\kappa'}|} \sum_{x_k \in C_k} \sum_{x_{k'} \in C_{k'}} d(x_k, x_{k'}) \tag{1}$$

can then be used to construct the agglomerative dendrogram that is used in this paper. There are various methods for determining the optimal number of clusters. Based on the Gap statistic and also the Silhouette methods using partitioning around medoids (PAMs) (Saunders et al. [21]), we obtained two clusters, but we decided to use three clusters based on the discussion in Section 2. This could help us better understand the extremal dependencies in those regions (clusters). Having more clusters results in some clusters having very few grid points which causes problems fitting the models (Saunders et al. [21]).

### 3.2. Max-Stable Processes

Max-stable processes form a class of processes when sample maxima are observed at each site of a spatial process, which is a problem of particular interest in connection with regional estimation methods in meteorology (Smith [23]). A max-stable process $Z(\cdot)$ is the limit process of maxima of independent identically distributed random fields $Y_i(x)$, $x \in \mathbb{R}^d$. For a pair of suitable normalising constants $a_n(x) > 0$ and $b_n(x) \in \mathbb{R}$, it then follows that (Smith [23])

$$Z(x) = \lim_{n \to \infty} \frac{\max_{i=1}^n Y_i(x) - b_n(x)}{a_n(x)}, \quad x \in \mathbb{R}^d. \tag{2}$$

The max-stable processes model in Equation (2) can be used in modelling annual maxima of spatial data. Various marginal transformations are usually preferred in fitting

spatial extremes models such as the max-stable processes. Amongst others, the unit Frèchet marginal transformation is given with Equation (3).

$$\Pr[Z(x) \leq z] = \exp\left(-\frac{1}{z}\right), \quad \forall x \in \mathbb{R}^d, \quad z > 0 \tag{3}$$

usually gains preference. In this study, the unit Frèchet marginal transformation is used. The first assumption is that the unit Frèchet marginal transformation holds, for which we have $a_n(x) = n$ and $b_n(x) = 0$. Smith [23] established the first class of max-stable process models, called the rainfall–storm model. More recently, Schlather [24] introduced a new characterisation of a max-stable process that allows for a random shape. Several max-stable models that might be relevant in modelling weather and climate extremes are discussed in this study.

### 3.3. The Smith Model

The characterisation of max-stable processes was initially proposed by Smith [23] and later used by Schlather [24] to provide a parametric model for spatial extremes. The construction of the Smith model is as follows: Let $\{(\xi_i, y_i), i \geq 1\}$ denote the points of a Poisson process on $(0 + \infty) \times \mathbb{R}^d$ with intensity measure $\xi^{-2}d\xi\nu(dy)$, where $\nu(dy)$ is a positive measure on $\mathbb{R}^d$. The characterisation of a max-stable process with unit Frèchet margins is given in Equation (4) (Smith [23]).

$$Z(x) = \max_i\{\xi_i f(y_i, x)\}, \quad x \in \mathbb{R}^d, \tag{4}$$

where $\{f(y, x), x, y \in \mathbb{R}^d\}$ is a nonnegative function such that

$$\int_{\mathbb{R}^d} f(x, y)\nu(dy) = 1, \quad \forall x \in \mathbb{R}^d. \tag{5}$$

To see that Equation (4) defines a stationary max-stable process with unit Frèchet margins, we have to check that the margins are indeed unit Frèchet $Z(x)$ and have the max-stable property. Following Smith [23], consider the set defined by the following:

$$E = \{(\xi, y) \in \mathbb{R}_*^+ \times \mathbb{R}^d : \xi f(y, x) > z\},$$

for a fixed location $x \in \mathbb{R}^d$ and $z > 0$. Then,

$$\Pr[Z(x) \leq z] \quad = \Pr[\text{no points in} \quad E] = \exp\left[-\int_{\mathbb{R}^d}\int_{z/f(y,z)} \xi^{-2}d\xi\nu(dy)\right]$$

$$= \exp\left[-\int_{\mathbb{R}^d} z^{-1}f(x, y)\nu(dy)\right] = \exp\left(-\frac{1}{z}\right),$$

where the margins are unit Frèchet. It then follows that

$$\left\{\max_{i=1}^n Z_i(x_1), \ldots, \max_{i=1}^n Z_i(x_k)\right\} \sim n\{Z(x_1), \ldots, Z(x_k)\}, \quad k \in \mathbb{N}.$$

The Smith model uses the Gaussian covariance function.

### 3.4. The Schlather Model

The second characterisation of the max-stable process was introduced by Schlather [24] as follows: Let $Y(\cdot)$ be a stationary process on $\mathbb{R}^d$ such that $\mathbb{E}[\max\{0, Y(x)\}] = 1$ and $\{\xi_i, i \geq 1\}$ are the points of a Poisson process on $\mathbb{R}_*^+$ with the intensity measurer $\xi^{-2}d\xi$. It

is established in Schlather [24] that a stationary max-stable process with the unit Frèchet margins can be defined with the following:

$$Z(x) = \max_i \xi_i \max\{0, Y_i(x)\},$$ (6)

where the $Y_i(\cdot)$ are the independent and identically distributed (i.i.d) copies of $Y(\cdot)$. The max-stable property of $Z(\cdot)$ stems from the superposition of $n$ independent identical Poisson processes, while the unit Frèchet margins hold the same argument as that for Smith [23]. Considering the set

$$E = \{(\xi, y(x)) \in \mathbb{R}_*^+ \times \mathbb{R}^d : \xi \max(0, y(x)) > z\}$$

for a fixed location $x \in \mathbb{R}^d$ and $z > 0$, then

$$\Pr[Z(x) \leq z] \quad = \Pr[\text{no points in E}] = \exp\left[-\int_{\mathbb{R}^d}\int_{z/\max(0,y(x))}^{+\infty} \xi^{-2} d\xi \nu(dy(x))\right]$$

$$= \exp\left[-\int_{\mathbb{R}^d} z^{-1}\max\{0,y(x)\}\nu(xy(x)))\right] = \exp\left(-\frac{1}{z}\right).$$

As with the Smith Model, the process defined in Equation (6) has a practical interpretation. For example, considering $\xi_i Y_i(\cdot)$ as the daily spatial rainfall events, all these events have the same spatial dependence structure but differ only in magnitude $\xi_i$ (Schlather [24]). This model differs slightly from the model in Smith [23], as we now have no deterministic shape, such as a multivariate normal density for the storms, but rather a random shape driven by the process $Y(\cdot)$ (Schlather [24]). To observe the connection between the Smith and the Schlather characterisations, we consider the case for which $Y_i(x) = f_0(x - X_i)$ where $f_0$ is a probability density function and $\{X_i\}$ is a homogeneous Poisson process both in $\mathbb{R}^d$.

With this particular setting, the models defined in Equations (4) and (6) are similar, as mentioned in Schlather [24]). However, Schlather [24] proposed to take $Y_i(\cdot)$ to be a stationary standard Gaussian process with correlation function $\rho(h)$, scaled so that $\mathbb{E}[\max\{0, Y_i(x)\}] = 1$. This is proposed since Equation (6) seems quite general, with the need for additional assumptions to obtain practical models. Based on these assumptions, it can be shown that the bivariate cumulative distribution function (CDF) of process (6) is as follows:

$$\Pr[Z(x_1) \leq z_1, Z(x_2) \leq z_2] = \exp\left[-\frac{1}{2}\left(\frac{1}{z_1} + \frac{1}{z_2}\right)\left(1 + \sqrt{1 - 2(\rho(h)+1)\frac{z_1 z_2}{(z_1+z_2)^2}}\right)\right],$$ (7)

where $h \in \mathbb{R}^+$ is the euclidean distance between location 1 and location 2. Usually, $\rho(h)$ is chosen from one of the valid parametric families of covariance functions, such as the power exponential, Whittle–Matern, Cauchy, and generalised Cauchy, among others.

### 3.5. Spatial Dependence and Extremal Coefficient of Max-Stable Processes

This paper is focused on understanding how dependence evolves in space. However, when dealing with non-extremal data, a common tool is the semivariogram $\gamma$, which may not be relevant here since this paper centers on the simultaneous occurrence of extremal events. This problem leads to a need for the development and use of more suitable tools for analysing the spatial dependence of max-stable fields. The extremal coefficient- and the variogram-based approaches that are especially well adapted to extremes are presented in this section as measures of the degree of dependence for extreme values.

Let $Z(\cdot)$ be a stationary max-stable random field with unit Frèchet margins. The extremal dependence among $N$ fixed locations in $\mathbb{R}^d$ can be summarised by the extremal coefficient, which is defined as follows:

$$\Pr[Z(x_1) \leq x, \dots, Z(x_N) \leq x] = \exp\left(-\frac{\theta_N}{z}\right), \tag{8}$$

where $1 \leq \theta_N \leq N$ with the lower and upper bounds correspond to complete dependence and independence, thus providing a measure of spatial dependence between stations. Given the properties of the max-stable process with unit Frèchet margins, the finite-dimensional CDF belongs to the class of multivariate extreme value distributions

$$\Pr[Z(x_1) \leq z_1, \dots, Z(x_N) \leq z_N] = \exp\{-V(z_1, \dots, z_N)\}, \tag{9}$$

where $V$ is a homogeneous function of order $-1$ that is referred to as the exponent measure. Consequently, the homogeneity property of $V$ implies a strong relationship between the exponent measure and the extremal coefficient

$$\theta_N = V(1, \dots, 1). \tag{10}$$

However, an important special case of the extremal coefficient defined in Equation (8) is to consider pairwise extremal coefficients given as

$$\Pr[Z(x_1) \leq z, Z(x_2) \leq z] = \exp\left\{-\frac{\theta(x_1 - x_2)}{z}\right\}. \tag{11}$$

Following Schlather and Tawn [25], $\theta(\cdot)$ is called the extremal coefficient function, which provides sufficient information about extremal dependence for many problems although it does not characterise the full distribution. With $z_1 = z_2 = z$, the extremal coefficient functions for max-stable models can be derived directly from their bivariate distribution. We then have the following:

$$\text{Smith} \qquad \theta(x_1 - x_2) = 2\Phi\left(\frac{\sqrt{(x_1-x_2)^T \Sigma^{-1}(x_1-x_2)}}{2}\right)$$

$$\text{Schlather} \qquad \theta(x_1 - x_2) = 1 + \sqrt{\frac{1-\rho(x_1-x_2)}{2}},$$

where $\Phi$ is the distribution function of the standard normal distribution, $\sum^{-1}$ denotes the covariance function and $\rho(.)$ is the correlation function.

### 3.6. $F-$Madogram

The use of the semivariogram and ordinary madogram is not appropriate for characterising dependence features of extreme variables. It is proposed in the literature that a modified madogram called the $F-$madogram should be used, which overcomes the limitation of both the semivariogram and ordinary madogram. The $F-$madogram is given in Equation (12).

$$\nu_F(x_1 - x_2) = \frac{1}{2}\mathbb{E}[|F(Z(x_1)) - F(Z(x_2))|] \tag{12}$$

where $Z(\cdot)$ is a stationary max-stable random field with unit Frèchet margins and $F(z) = \exp(-1/z)$ (Ribatet [26]). In the presence of the Frèchet choice of marginal transformation, Equation (12) suggests a simple estimator given with Equation (13).

$$\hat{\nu}_F(x_1 - x_2) = \frac{1}{2n}\sum_{i=1}^{n}|\hat{F}(z_i(x_1)) - \hat{F}(z_i(x_2))|, \tag{13}$$

where $z_i(x_1)$ and $z_i(x_2)$ are the $i$th observations of the random field at locations $x_1$ and $x_2$ and $n$ is the total number of observations. The strong connection between the $F-$madogram and the extremal coefficient is given in Equation (14):

$$2\nu_F(x_1 - x_2) = \frac{\theta(x_{-}x_2) - 1}{\theta(x_1 - x_2) + 1}. \tag{14}$$

It then follows from Equation (14) that there is a one-to-one relationship between the extremal coefficient and the $F-$madogram which suggests a simple estimator for $\theta(x_1 - x_2)$ given by Equation (15).

$$\hat{\theta}(x_1 - x_2) = \frac{1 + 2\hat{\nu}_F(x_1 - x_2)}{1 - 2\hat{\nu}_F(x_1 - x_2)}. \tag{15}$$

*3.7. Parametric Inference: Maximum Pairwise Likelihood Estimation*

The full likelihood function for the max-stable model is not analytically known if the number of stations or grid points under consideration is at least three (Ribatet [26]). Since the bivariate density is analytically known, this suggests using the pairwise likelihood instead of the full likelihood. The parametric inference in this paper entails maximising the pairwise likelihood instead of the full likelihood. The log pairwise likelihood is given with Equation (16):

$$\ell_\rho(\mathbf{Z}, \psi) = \sum_{i < j} \sum_{k=1}^{n_{i,j}} \log f\big(Z_k^{(i)}, Z_k^{(j)}; \psi\big), \tag{16}$$

where $\mathbf{Z}$ constitutes the data available on the whole region, $n_{i,j}$ is the number of common observations between grid points $i$ and $j$, $y_k^{(i)}$ is the $k$th observation of the $i$th grid and $f(\cdot, \cdot)$ is the bivariate density of the unit Fréchet max-stable process. Since the maximum pairwise likelihood estimation (MPLE) $\hat{\psi}_\rho$ has the same properties as those of the maximum likelihood estimation (MLE), it then follows that

$$\hat{\psi}_\rho \dot\sim N\big(\psi, H(\psi)^{-1} J(\psi) H(\psi)^{-1}\big),$$

where $H(\psi) = \mathbb{E}[\nabla^2 \ell_\rho(\psi; \mathbf{Z})]$ and $J(\psi) = \text{Var}[\nabla \ell_\rho(\psi; \mathbf{Z})]$, and the expectations are with respect to the full density. In the presence of misspecification, using the Akaike Information Criterion (AIC) (Akaike [27]) is not justified as the second Bartlett identity is not satisfied. This implies the following:

$$\mathbb{E}[\nabla^2 \ell_\rho(\psi; \mathbf{Z})] + \text{Var}[\nabla \ell_\rho(\psi; \mathbf{Z})] \neq 0.$$

The max-stable processes in this paper are misspecified, and using the Takeuchi Information Criterion (TIC) (Takeuchi [28]) is preferable. This is given with the following:

$$\text{TIC} = -2\ell_\rho(\hat{\psi}_\rho) - 2\text{tr}\big\{J(\hat{\psi}_\rho) H(\hat{\psi}_\rho)^{-1}\big\},$$

where $\ell(.)$ denotes the log likelihood function, $J$ is the Jacobian and $H$ the hat matrix.

## 4. Empirical Results

*4.1. Exploratory Data Analysis and Hierarchical Clustering*

As part of the exploratory analysis in this study, we deemed it vital to understand the pairwise relationship of the variables under consideration. The temperature and precipitation data distributions are visualised using histograms and a pairwise scatterplot (Figure 5). The pairwise Kendall's rank correlation coefficient, which is 0.37 in Figure 5, shows some positive correlation between temperature and precipitation in the study area. However, there is a decrease in rainfall for high temperatures.

Six of the twenty-four grid locations were selected, and plots of precipitation and temperature are given in Figures 6 and 7, respectively. Visual inspection of Figure 6 suggests

for all the plots that the data are stationary. However, there appears to be an upward trend in temperature in all the plots of Figure 7. The grid locations are denoted by $r_ic_j$, where $i$ represents latitudes, and $j$ denotes longitudes. A summary of coordinates (latitudes and longitudes) of the 24 grid locations is given in Appendix A.1.

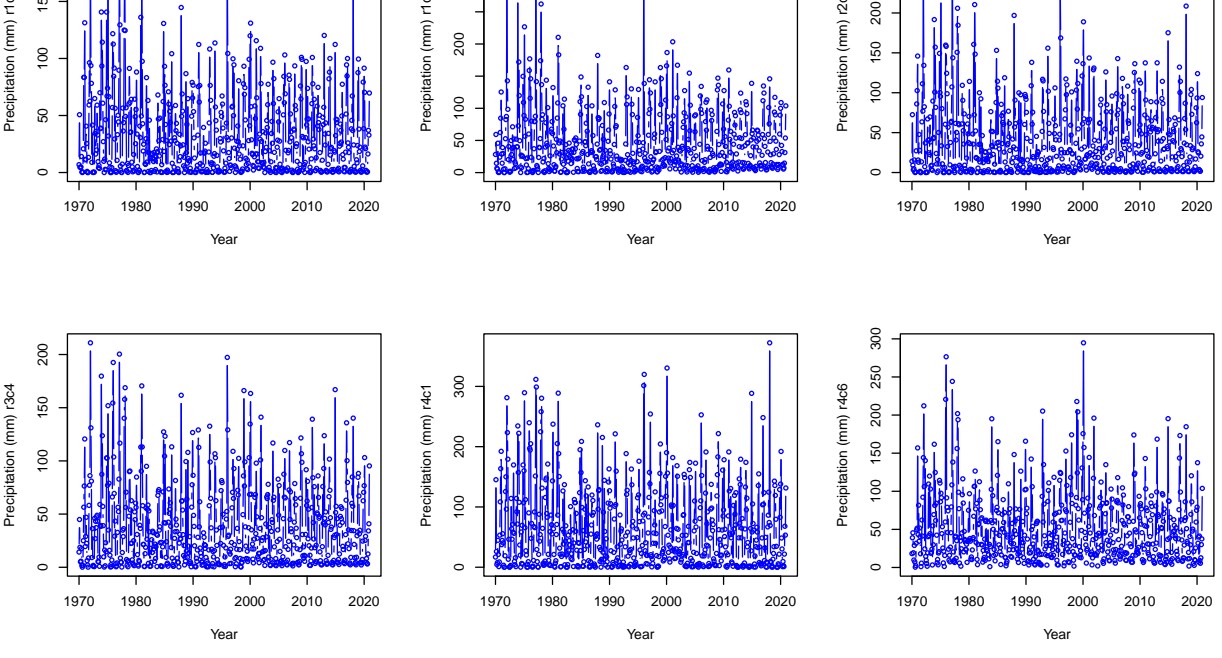

**Figure 5.** Histograms (diagonal), pairwise scatter plots (bottom left), and pairwise Kendall's rank correlation coefficient (top right) of temperature and rainfall data for all the study area.

**Figure 6.** Plots of precipitation for the grids: r1c1, r1c5, r2c3, r3c4, r4c1 and r4c6.

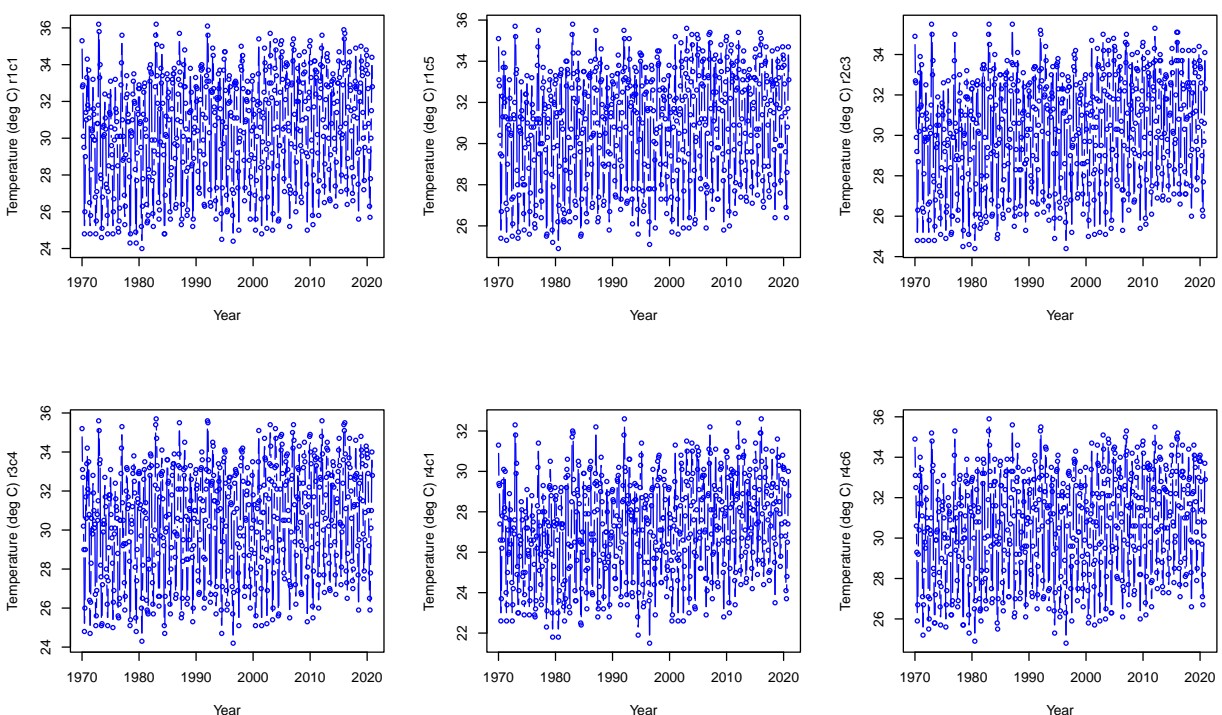

**Figure 7.** Plots of temperature for the grids: r1c1, r1c5, r2c3, r3c4, r4c1 and r4c6.

The Mann–Kendall test (Mann [29], Kendall [30]) was used to check for the existence of monotonic trends in both precipitation and temperature from six selected grid locations. The null hypothesis of no monotonic trend in all six locations was accepted since the *p*-values were all greater than the 0.05 level of significance (Table 1). We then used Sen's slope (Sen [31]) method to compute the magnitude of the slopes. Based on the *p*-values, the magnitudes of the computed slopes were all insignificant (Table 1).

**Table 1.** Trend testing for selected grid points (precipitation).

| | **Mann–Kendall Trend Test** | | | | | |
|---|---|---|---|---|---|---|
| | **r1c1** | **r1c5** | **r2c3** | **r3c4** | **r4c1** | **r4c6** |
| *p*-value | 0.484 | 0.671 | 0.947 | 0.985 | 0.130 | 0.505 |
| | **Magnitude of Trend** | | | | | |
| *p*-value | 0.484 | 0.671 | 0.947 | 0.985 | 0.130 | 0.505 |
| Sen's slope | −0.0017 | 0.0022 | 0 | 0 | −0.0090 | −0.0047 |

The same tests, Mann–Kendall and Sen's test, were carried out on the temperature data on the selected six grid locations. Results (see Table 2) suggest significantly increasing trends in all cases.

Before the clustering methods and algorithms can be applied to the relevant data, the first step is to assess whether it is reasonable to cluster that detailed data (Adolfsson et al. [32]). This process is called the assessment of clustering tendency and is considered important for avoiding misleading clustering results. In this paper, the assessment of clustering tendency was performed using the Hopkins statistic introduced by Hopkins and Skellam [33]. According to Hopkins and Skellam [33], if the value of the Hopkins statistic is close to 1 (far above 0.5), it can be concluded that the dataset is significantly clusterable. In this paper, Hopkin's statistic for annual average precipitation and Hopkin's statistic for

annual maximum temperature is found to be 0.7317 and 0.8446, respectively. These values confirm that the precipitation and temperature data can be reasonably clustered.

**Table 2.** Trend testing for selected grid points (temperature).

| | **Mann–Kendall Trend Test** | | | | | |
| | **r1c1** | **r1c5** | **r2c3** | **r3c4** | **r4c1** | **r4c6** |
|---|---|---|---|---|---|---|
| *p*-value | $1.47 \times 10^{-5}$ | $3.94 \times 10^{-5}$ | $1.66 \times 10^{-5}$ | $4.62 \times 10^{-5}$ | $2.62 \times 10^{-7}$ | $3.42 \times 10^{-5}$ |
| | **Magnitude of Trend** | | | | | |
| *p*-value | $1.47 \times 10^{-5}$ | $3.94 \times 10^{-5}$ | $1.66 \times 10^{-5}$ | $4.62 \times 10^{-5}$ | $2.62 \times 10^{-7}$ | $3.42 \times 10^{-5}$ |
| Sen's slope | 0.0030 | 0.0025 | 0.0028 | 0.0026 | 0.0030 | 0.0025 |

Saunders et al. [21] define a cluster dendrogram as a diagram that shows the hierarchical relationship between objects. Each point is considered its cluster or leaf in the dendrogram, whereas the branches in the dendrogram are formed by successively combining leaves and other branches until all points are grouped. We considered three clusters for annual average precipitation and three for annual maximum temperature in clustering the study area. This was based on the discussion presented in Section 2. The dendrogram for the annual average precipitation and the dendrogram for the annual maximum temperatures are respectively shown at the top-left panel and the top-right panels of Figure 8. The number of observations in clusters one, two and three are 8, 10 and 6, respectively. Similarly, for temperature data, clusters one, two and three have 3, 16 and 5 observations, respectively. According to Saunders et al. [21], clustering is more appropriate when the number of observations in the cluster is relatively large. The maps for the annual average precipitation and the annual maximum temperature in the Lowveld region of Limpopo with three clusters are given in Appendix A.3 in Figures A1 and A2, respectively.

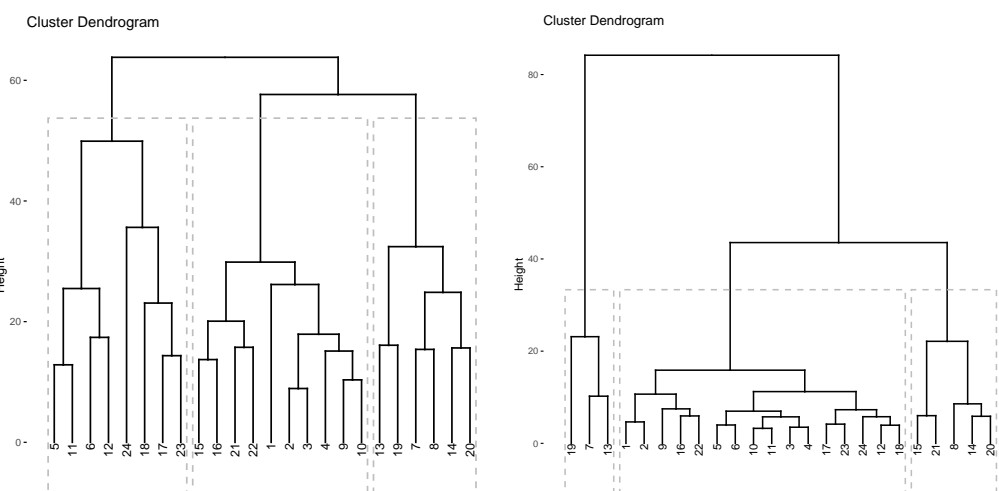

**Figure 8.** *Cont.*

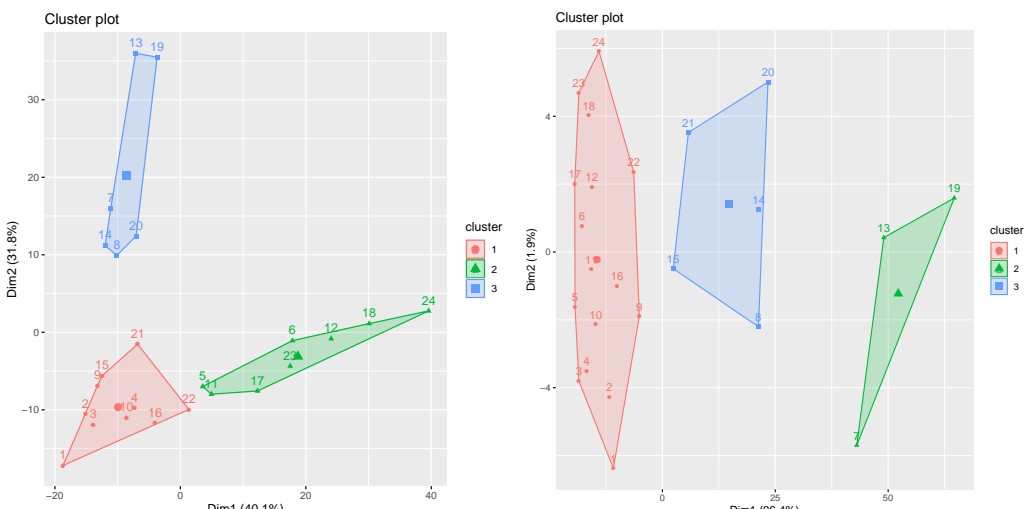

**Figure 8.** (**Top-left panel**) Cluster dendrogram for precipitation data. (**Top-right panel**) Cluster dendrogram for temperature data. (**Bottom-left panel**) cluster scatter plots for precipitation data. (**Bottom-right panel**) cluster scatter plots for temperature data.

### 4.2. Empirical Results from Max-Stable Process

The Smith and Schlather max-stable process models were fitted to each cluster of annual average precipitation. The results are shown in Tables 3, 4 and 5, respectively. The results for fitting the two families of max-stable models to the annual maximum temperature are summarised in Tables 6, 7 and 8, respectively. These tables present, among other data, the TIC and the parametric estimates. The smallest value of the TIC was used for choosing the best out of various nested models. Looking at cluster 1 of the precipitation data in Table 3, the Schlather model with the powered exponential covariance function is the best model based on TIC =198,585.3. Similarly, the best fitting model for clusters 2 and 3 is the Schlather with the powered exponential and Whittle–Matern covariance functions, respectively.

**Table 3.** Max-stable process fitted to the first cluster of the average annual precipitation.

| Model | Covariance Family | TIC | Deviance | Range | Smooth | Smooth2 | Cov11 | Cov12 | Cov22 |
|---|---|---|---|---|---|---|---|---|---|
| Schlather | Powered Exponential | 198,585.3 | 198,558.2 | 3.68 (0.27) | 1.31 (0.04) | | | | |
| Schlather | Whittle-Matern | 198,586.7 | 198,559.7 | 2.88 (0.32) | 0.75 (0.05) | | | | |
| Schlather | Cauchy | 198,608.5 | 198,581.4 | 0.74 (0.05) | 0.19 (0.02) | | | | |
| Schlather | Generalized Cauchy | 198,589 | 198,560.4 | 5.35 (9.29) | 2.61 (5.24) | 1.37 (0.11) | | | |
| Smith | Gaussian | 203,737.5 | 203,714.5 | | | | 1.04 (0.02) | 0.38 (0.01) | 1.28 (0.02) |

**Table 4.** Max-stable process fitted to the second cluster of the average annual precipitation.

| Model | Covariance Family | TIC | Deviance | Range | Smooth | Smooth2 | Cov11 | Cov12 | Cov22 |
|---|---|---|---|---|---|---|---|---|---|
| Schlather | Powered Exponential | 123,866.9 | 123,842.6 | 2.13 (0.15) | 1.74 (0.07) | | | | |
| Schlather | Whittle-Matern | 123,895.4 | 123,871.1 | 0.10 (1.46) | 95.72 (2906.14) | | | | |
| Schlather | Cauchy | 123,895.4 | 123,871.1 | 18.22 (289.8) | 97.22 (3092.2) | | | | |
| Schlather | Generalized Cauchy | 123,869 | 123,844.1 | 1.55 (0.58) | 1.56 (0.87) | 1.98 (0.11) | | | |
| Smith | Gaussian | 125,686.3 | 125,667.8 | | | | 0.91 (0.02) | 0.54 (0.02) | 1.28 (0.03) |

**Table 5.** Max-stable process fitted to the third cluster of the average annual precipitation.

| Model | Covariance Family | TIC | Deviance | Range | Smooth | Smooth2 | Cov11 | Cov12 | Cov22 |
|---|---|---|---|---|---|---|---|---|---|
| Schlather | Powered Exponential | 64,483.66 | 64,471.07 | 3.49 (0.38) | 1.29 (0.07) | | | | |
| Schlather | Whittle-Matern | 64,483.17 | 64,470.61 | 2.81 (0.45) | 0.73 (0.06) | | | | |
| Schlather | Cauchy | 64,479.89 | 64,467.29 | 0.57 (0.06) | 0.14 (0.02) | | | | |
| Schlather | Generalized Cauchy | 64,480.68 | 64,467.29 | 0.57 (0.39) | 0.28 (0.17) | 2.0 (0.73) | | | |
| Smith | Gaussian | 66,704.63 | 66,691.77 | | | | 0.65 (0.02) | −0.05 (0.02) | 0.86 (0.02) |

**Table 6.** Max-stable process fitted to the first cluster of the annual maximum temperatures.

| Model | Covariance Family | TIC | Deviance | Range | Smooth | Smooth2 | Cov11 | Cov12 | Cov22 |
|---|---|---|---|---|---|---|---|---|---|
| Schlather | Powered Exponential | 317,743.3 | 317,625.6 | 61.12 (6.69) | 1.25 ( 0.03) | | | | |
| Schlather | Whittle-Matern | 317,740.5 | 317,622.8 | 60.73 (6.79) | 0.64 (0.02) | | | | |
| Schlather | Cauchy | 317,699.6 | 317,581.8 | 0.77 (0.03) | 0.01 (0.00) | | | | |
| Schlather | Generalized Cauchy | 198,589 | 198,560.4 | 5.35 (9.29) | 2.61 (5.24) | 1.37 (0.11) | | | |
| Smith | Gaussian | 348,786 | 348,699.7 | | | | 51.54 (0.46) | −7.54 (0.25) | 21.80 (0.32) |

**Table 7.** Max-stable process fitted to the second cluster of the annual maximum temperatures.

| Model | Covariance Family | TIC | Deviance | Range | Smooth | Smooth2 | Cov11 | Cov12 | Cov22 |
|---|---|---|---|---|---|---|---|---|---|
| Schlather | Powered Exponential | 8292.54 | 8288.27 | 26.48 (7.58) | 1.26 (0.09) | | | | |
| Schlather | Whittle-Matern | 8292.54 | 8288.27 | 26.03 (7.85) | 0.65 (0.05) | | | | |
| Schlather | Cauchy | 8292.54 | 8288.27 | 0.54 (0.07) | 0.01 (0.00) | | | | |
| Schlather | Generalized Cauchy | N/A | 8288.27 | 4.37 (N/A) | 0.16 (N/A) | 1.32 (1.32) | | | |
| Smith | Gaussian | N/A | 8844.83 | | | | 43.03 (N/A) | 1.79 (N/A) | 11.12 (N/A) |

**Table 8.** Max-stable process fitted to the third cluster of the annual maximum temperatures.

| Model | Covariance Family | TIC | Deviance | Range | Smooth | Smooth2 | Cov11 | Cov12 | Cov22 |
|---|---|---|---|---|---|---|---|---|---|
| Schlather | Powered Exponential | 25,147.82 | 25,137.75 | 232.46 (123.98) | 0.88 (0.08) | | | | |
| Schlather | Whittle-Matern | 25,147.81 | 25,137.75 | 229.28 (119.98) | 0.44 (0.04) | | | | |
| Schlather | Cauchy | 25,149.51 | 25,139.5 | 0.28 (0.04) | 0.00 (0.00) | | | | |
| Schlather | Generalized Cauchy | 25,148.66 | 25,137.86 | 8.32 (147.62) | 0.06 (0.82) | 0.93 (0.62) | | | |
| Smith | Gaussian | 27,471.18 | 27,461.64 | | | | 21.30 (0.20) | −9.30 (0.13) | 17.37 (0.193) |

Likewise, based on the results given in Table 6, the best fitting models to the annual maximum temperature are the Schlather with the generalised Cauchy, Cauchy and Whittle–Matern covariance functions for clusters 1, 2 and 3, respectively. The diagnostic plots for the max-stable process models fitted to each cluster of annual average precipitation, and each cluster of annual maximum temperature are shown in Figures 9 and 10, respectively. Figures 9 and 10, respectively, show plots of the first column extremal-coefficient $\theta(h)$ against the distance $h$ for the average annual precipitation and the annual maximum temperature. From Figure 9, column one, the empirical point clouds of cluster 1 show a fairly large variability of $\theta$ around the mean value. For example, in station cluster 1, the distance of $h = 0.5$ has a range of very different values of $\theta$, spanning from 1.09 to 1.24. However, the variability is low for the temperature data, as shown in the first column of the empirical point clouds for all three clusters as shown in Figure 10. Plots of the extremal coefficients as the longitudes and latitudes change are given in the right panels of Figures 9 and 10, respectively.

Other pairs of the diagnostic plots for the max-stable process models are shown in Appendix A.3. Figure A3 shows the fit of the max-stable process model fitted to the first cluster of average annual precipitation with a powered exponential covariance family. This pair of diagnostic plots confirm a good fit of the max-stable process model with a powered exponential covariance family. Focusing on the annual maximum temperature, the diagnostic plots in Figure A4 show the fit of the max-stable process model fitted to the first cluster of annual maximum temperature with a generalised Cauchy covariance

family. These also confirm the good fit of the max-stable process to the annual maximum temperature. The red line in the diagnostic plots shows the theoretical extremal coefficient, whereas the grey points are pairwise estimates, and black crosses are binned estimates.

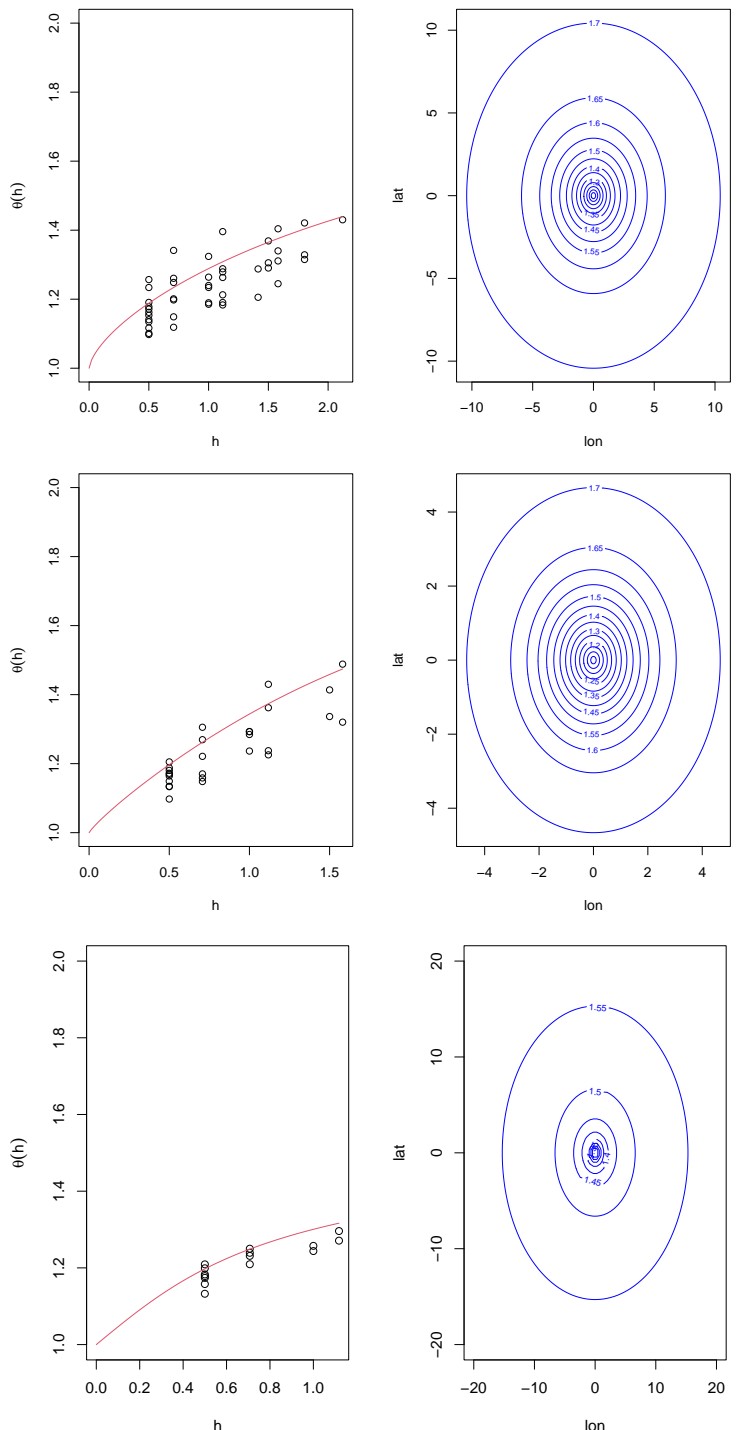

**Figure 9.** (**Top panel**) Cluster 1, (**Middle panel**), cluster 2 and (**Bottom panel**) Cluster 3 plots of (i) first column extremal coefficient $\theta(h)$ against distance $h$ for the average annual precipitation and (ii) second column extremal coefficient changes as longitudes and latitudes evolve.

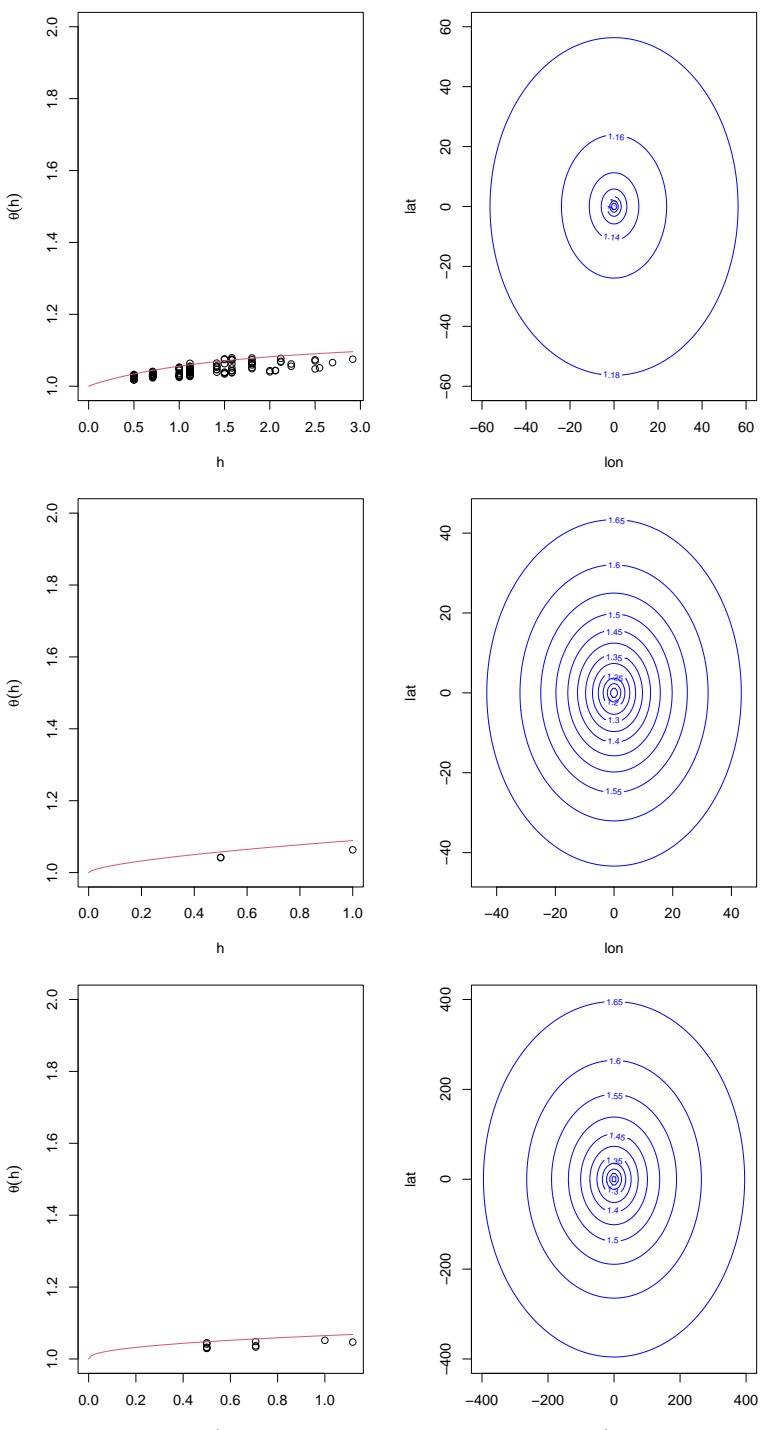

**Figure 10.** (**Top panel**) Cluster 1, (**Middle panel**), cluster 2 and (**Bottom panel**) cluster 3 plots of (i) first column extremal coefficient $\theta(h)$ against distance $h$ for the maximum annual temperature and (ii) second column extremal coefficient changes as longitudes and latitudes evolve.

The estimation of the extremal index for the annual average precipitation and annual maximum temperature is given in Tables 9 and 10, respectively. Table 9 shows that the extremal dependence gets weaker as distance increases. This suggests that heavy rainfall in one station in a cluster will not result in heavy rainfall at another location in the same

cluster. This could be a result of the different topographic features in the cluster (region), which are known to influence rainfall strongly (Saunders et al. [21]).

**Table 9.** Estimation of the extremal index for the precipitation data.

|  | $h$ | 0.5 | 1.0 | 1.5 | 2.0 |
|---|---|---|---|---|---|
| Cluster 1 | $\theta(h)$ | 1.14 | 1.22 | 1.32 | 1.44 |
| Cluster 2 | $\theta(h)$ | 1.18 | 1.32 | 1.42 | N/A |
| Cluster 3 | $\theta(h)$ | 1.22 | 1.33 | N/A | N/A |

The extremal dependence for all three clusters for the temperature data is very strong even for larger values of distance $h$, as seen in Table 10. This suggests that if one station in each cluster is hot, then it is expected that the rest of the stations in that cluster will also experience hot weather. This modelling could be useful to meteorologists, including climatologists and other relevant stakeholders such as those in the agricultural sector.

**Table 10.** Estimation of the extremal index for the temperature data.

|  | $h$ | 0.5 | 1.0 | 1.5 | 2.0 |
|---|---|---|---|---|---|
| Cluster 1 | $\theta(h)$ | 1.02 | 1.03 | 1.05 | 1.06 |
| Cluster 2 | $\theta(h)$ | 1.04 | 1.06 | N/A | N/A |
| Cluster 3 | $\theta(h)$ | 1.04 | 1.05 | N/A | N/A |

## 5. Discussion

The study used gridded data to analyse the extremal dependence of temperature, including precipitation data using two max-stable processes. To our knowledge, no such analysis has been done in the Lowveld of the Limpopo Province in South Africa.

The current study is an extension of the work done by Saunders et al. [21]. In contrast to the work by Saunders et al. [21], this study used gridded precipitation and temperature data. The agglomerative method of hierarchical clustering was used to cluster the study area. This method is known to improve cohesion, and the partitions have an interpretation of the strength of the dependence (Saunders et al. [21]). The silhouette and gap statistic were used in the study to determine the optimal number of clusters using partitioning around medoids (PAMs). Rainfall and temperature data were initially transformed to unit Frèchet margins before fitting the max-stable models. From the precipitation data, the Schlather model with exponential covariance function was the best fitting model for clusters 1 and 2, while the Schlather model with the Cauchy was found to be the best fitting model for cluster 3. Similarly, with temperature data, the Schlather model was found to be the best fitting model with the generalised Cauchy, the power exponential and the Whittle–Matern as the best covariance functions for clusters 1, 2 and 3, respectively.

A comparison of the extremal coefficients to their respective pairwise estimates for both rainfall and temperature data is given in Figures 9 and 10, respectively. The pairwise estimates are close to all cases' red curves (the theoretical extremal coefficients). From Figure 9 (rainfall data), it can be seen that as distance increases, the extremal dependence gets weaker for all three clusters. Unlike the rainfall data, the extremal dependence remains strong as distance increases, as seen in Figure 10. The results from the two figures were confirmed by estimates of the extremal coefficients summarised in Tables 9 and 10, respectively. The results show that the $F-$madogram provides partial and not full extremal dependence. As suggested by Huser and Genton [34] and Oesting et al. [35], where the location includes orography, a single dependence structure may not be appropriate.

For modelling the risk of drought in the study area, the region of interest is on longitudes $30.5°$ to $32.5°$ and latitudes $-22°$ to $-23.5°$. This area is covered by clusters 1 and 2 for precipitation and clusters 1 and 3 for temperature. See Figures A1 and A2. It is noted that for clusters 1 and 2 (precipitation), as $h$ increases, extremal dependence decreases significantly. For details, see Figure 9 (top and left middle panels) and Table 9

(clusters 1 and 2 rows). Similarly, extremal dependence is very strong for temperature in clusters 1 and 3, even for large values of $h$. See Figure 10 (top and bottom left panels) and Table 10 (clusters 1 and 3 rows). These findings are consistent with the comprehensive understanding of the climate of the study area and its topographic features (Chikoore [16]).

A summary of some of the research highlights is as follows:

- The agglomerative hierarchical clustering method was used to cluster the study area;
- The Smith and Schlather models were fitted to the clustered data in an attempt to understand the extremal dependence in each cluster;
- Three clusters were selected from each variable, i.e., temperature and precipitation data, before fitting the max-stable models;
- The Schlather model with the powered exponential covariance was found to be the best-fitting model for the precipitation data in clusters 1 and 2, respectively. The Schlather model with the Whittle–Matern outperformed the others in regard to the third cluster. As for temperature data, the Schlather model with the generalised Cauchy, Cauchy and Whittle–Matern were the best-fitting models for clusters 1, 2 and 3, respectively.

The modelling framework presented in this paper could be useful to climatologists and hydrologists, including decision-makers in the agricultural sector, in understanding the behaviour and impact of the risk of drought caused by high temperatures and low precipitation. Human activity interaction is also needed to reduce the potential impacts of compound precipitation and temperature extremes. This should include an in-depth understanding and analysis of the physical processes leading to compound extremes.

The limitations of this study include the fact that our application study did not cover the full range of dependence due to the difficulties in estimating the parameters for full dependence. To address this limitation, use was made of the $F-$madogram, which is only a measure of partial extremal dependence, and as such, we did not need to rely solely on quantifying extremal dependence based on the $F-$madogram. However, the empirical estimates suggest using a nugget effect that has to be linked to the theoretical interpretation. Further, it is worth noting that the study area is generally flat or gently undulating along the floors of the major river valleys, with average elevations of 450 m and decreasing elevations to the east.

## 6. Conclusions

Depending on the topography and climatic conditions, droughts vary from one region to another. In this paper, we presented an application of some spatial extremes models in assessing the risk of drought caused by the simultaneous occurrence of low precipitation rates and extremely high temperatures in the Lowveld region of Limpopo Province in South Africa. We considered using the CRU TS 4.05 gridded average precipitation and the maximum temperature at a 0.5-degree resolution. The agglomerative hierarchical clustering method was used to cluster the study area before fitting max-stable processes. This was done to understand the dependence features adequately.

**Author Contributions:** Conceptualisation, M.M.N. and C.S.; methodology, M.M.N. and C.S.; software, M.M.N.; validation, M.M.N., C.S., H.C. and A.B.; formal analysis, M.M.N.; investigation, M.M.N., C.S., H.C. and A.B.; data curation, M.M.N. and H.C.; writing—original draft preparation, M.M.N.; writing—review and editing, M.M.N., C.S., H.C. and A.B.; visualisation, M.M.N. and C.S.; supervision, C.S., H.C. and A.B.; project administration, C.S., H.C. and A.B. All authors have read and agreed to the published version of the manuscript.

**Funding:** This research received no external funding.

**Institutional Review Board Statement:** Not applicable.

**Informed Consent Statement:** Not applicable.

**Data Availability Statement:** This study used gridded precipitation and temperature data on a $-22°$ north to $-24°$ north and a longitude of approximately $30°$ east to $33°$ east. The data were collected

from the KNMI climate explorer https://climexp.knmi.nl/start.cgi (accessed on 31 October 2022). The analytic data can be downloaded from https://github.com/csigauke (accessed on 31 October 2022) .

**Acknowledgments:** The authors are grateful to the numerous people for their helpful comments on this paper. The authors would like to acknowledge the significant contribution of the late lead author Murendeni M Nemukula. May his soul rest in eternal peace.

**Conflicts of Interest:** The authors declare no conflicts of interest.

## Abbreviations

The following abbreviations are used in this manuscript:

| | |
|---|---|
| AIC | Akaike Information Criterion |
| CDF | Cumulative Distribution Function |
| CRU TS | Climatic Research Unit gridded Time Series |
| KNMI | The Royal Netherlands Meteorological Institute |
| MLE | Maximum Likelihood Estimation |
| MPLE | Maximum Pairwise Likelihood Estimation |
| PAM | Partitioning Around Mediods |
| TIC | Takeuchi Information Criterion |

## Appendix A. Metadata, Maps Showing Clusters of Precipitation and Temperature Including Diagnostic Plots

*Appendix A.1. Metadata*

**Table A1.** Metadata.

| Grid ID | r1c1 | r1c2 | r1c3 | r1c4 | r1c5 | r1c6 |
|---|---|---|---|---|---|---|
| Latitude | −22.5 | −22.5 | −22.5 | −22.5 | −22.5 | −22.5 |
| Longitude | 30 | 30.5 | 31 | 31.5 | 32 | 32.5 |
| **Grid ID** | **r2c1** | **r2c2** | **r2c3** | **r2c4** | **r2c5** | **r2c6** |
| Latitude | −23 | −23 | −23 | −23 | −23 | −23 |
| Longitude | 30 | 30.5 | 31 | 31.5 | 32 | 32.5 |
| **Grid ID** | **r3c1** | **r3c2** | **r3c3** | **r3c4** | **r3c5** | **r3c6** |
| Latitude | −23.5 | −23.5 | −23.5 | −23.5 | −23.5 | −23.5 |
| Longitude | 30 | 30.5 | 31 | 31.5 | 32 | 32.5 |
| **Grid ID** | **r4c1** | **r4c2** | **r4c3** | **r4c4** | **r4c5** | **r4c6** |
| Latitude | −24 | −24 | −24 | −24 | −24 | −24 |
| Longitude | 30 | 30.5 | 31 | 31.5 | 32 | 32.5 |

*Appendix A.2. Maps Showing Three Clusters for Precipitation and Temperature Data*

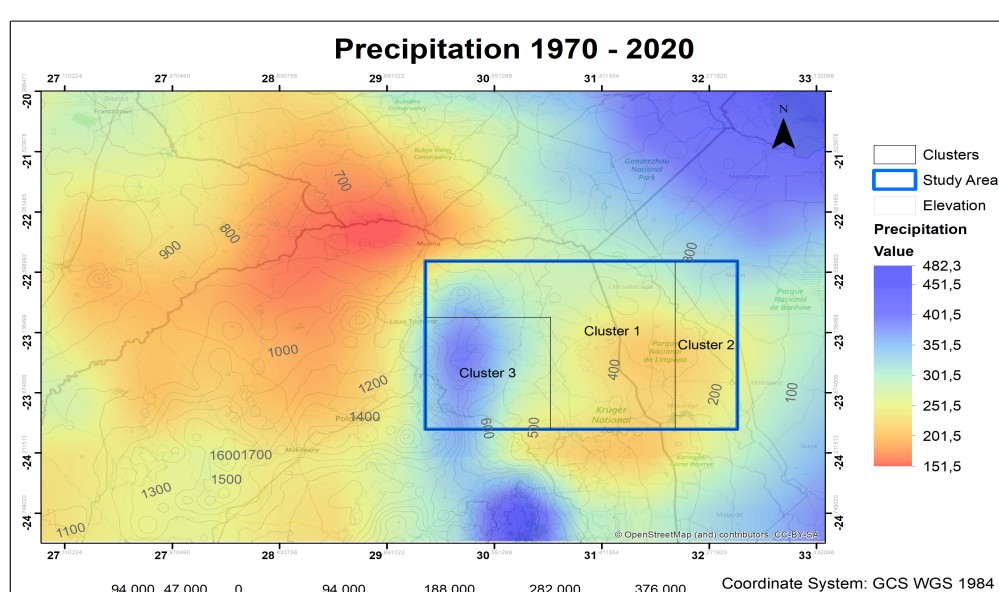

**Figure A1.** Map for the annual average precipitation in the Lowveld region of Limpopo with three clusters. Source: Authors' creation.

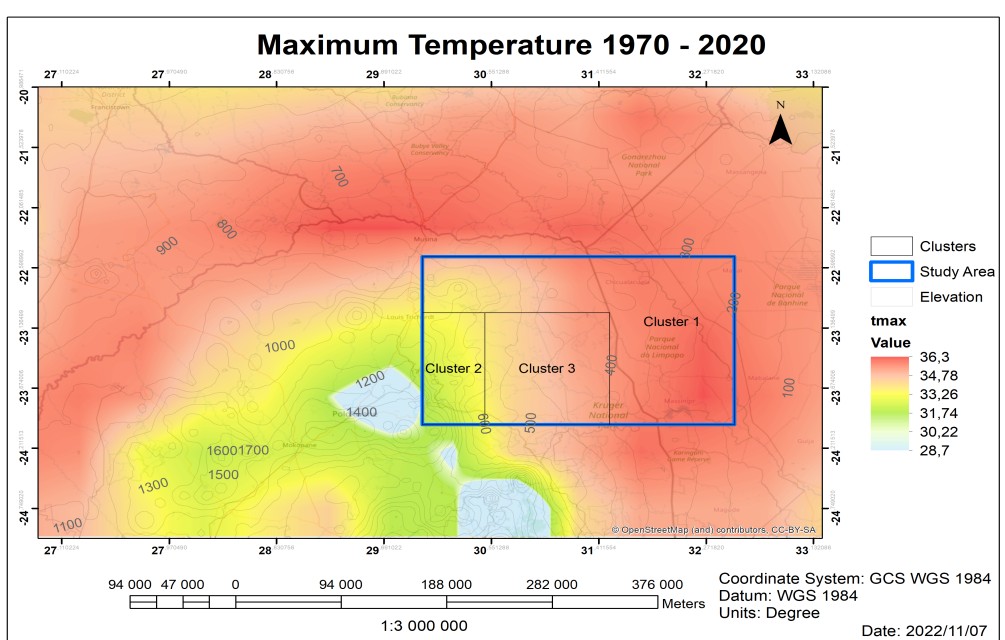

**Figure A2.** Map for the annual maximum temperature in the Lowveld region of Limpopo with three clusters. Source: Authors' creation.

*Appendix A.3. Diagnostic Plots for the Max-Stable Processes Fitted to Clusters of Precipitation and Temperature Data*

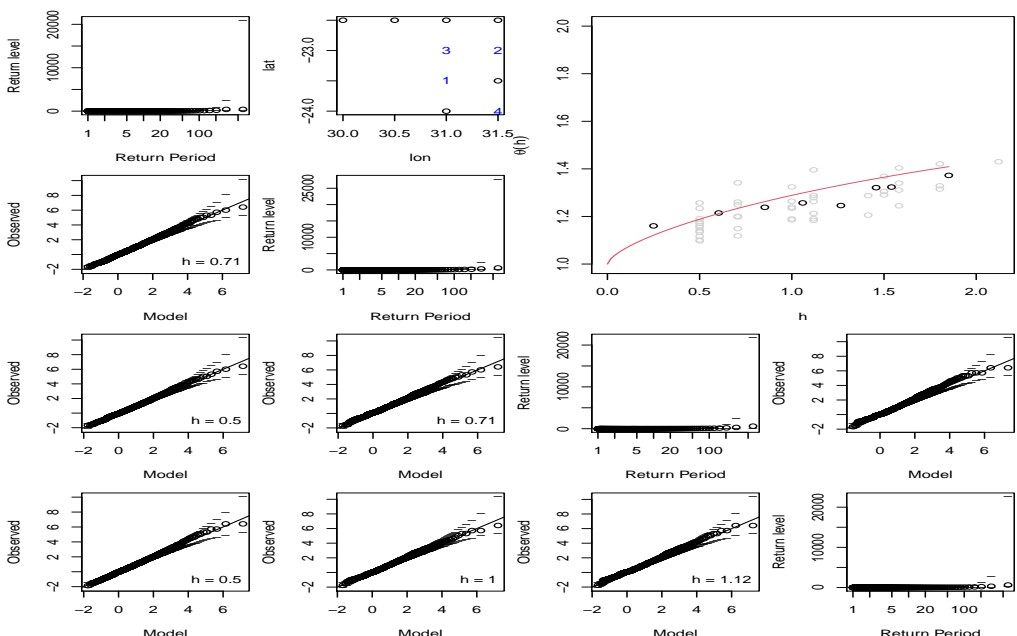

**Figure A3.** Diagnostic plots of the max-stable process fitted to the first cluster of average annual precipitation with a powered exponential covariance family.

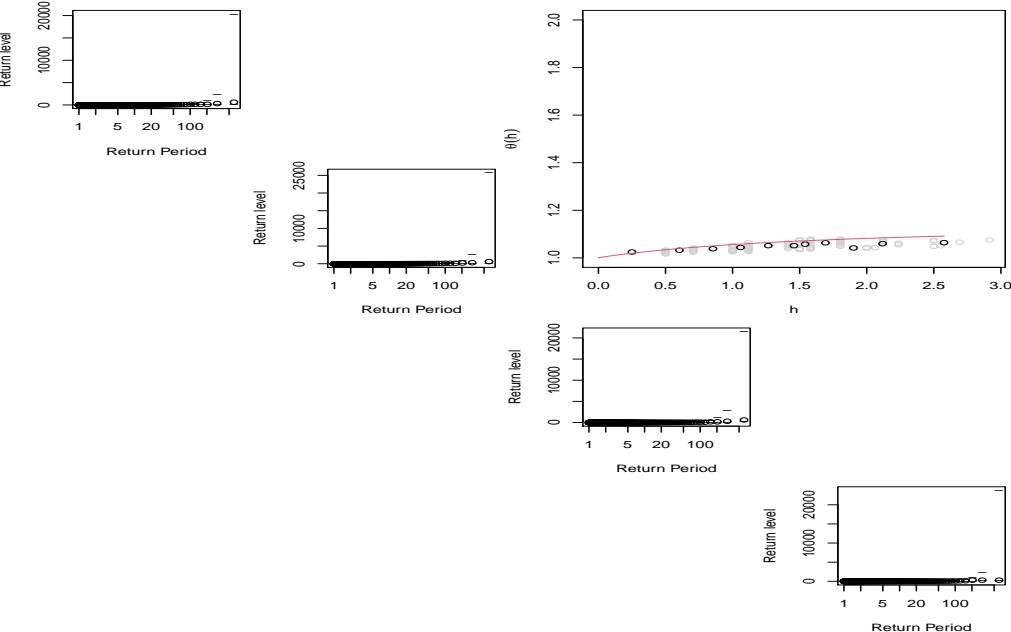

**Figure A4.** Diagnostic plots of the max-stable process fitted to the first cluster of annual maximum temperature with a generalised Cauchy covariance family.

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
