# Peer review of "Modelling Drought Risk Using Bivariate Spatial Extremes: Application to the Limpopo Lowveld Region of South Africa"

_climate, doi:10.3390/cli11020046_

Round 1
Reviewer 1 Report
Thanks to the author for providing an interesting paper on the simulation of drought risk using an extreme spatial multivariate approach. However, the author needs to recalibrate so that the reader can follow. Some suggestions.
1. Title: the author notes the drought risk. However, the results and the discussion section needed to be presented. Therefore, the author needs to reconsider the consistency between the content and the article title
2. Introduction: it needed to be longer. More literature review needs to be added
3. Method section: the author needs to rearrange the content more coherently. It may be a good idea to use a diagram to present the overall methods used in this study.
4. Results section: more concise in presenting the results. The drawings need to be represented carefully.
5. Discussion section: What does it mean to delve into such a result? Or why the results are more than just restating the results.
The most important thing is that the author should rearrange the article’s structure to be easily accessible to the reader.

Author Response
-
Thank you for your valuable comments, please check the attachment.

Reviewer 2 Report
1. re set the methodology flow chart. it is not correct, there is no check no correction only smooth flow to end.
2. references are not cited correctly check line 22, 26, 27 and 30. be consistent in reference style
3, cite new reference form 2022 and 2023
Author Response

(The authors gave the same response as above.)

Reviewer 3 Report
Authors have properly addressed all the raised concerns. now, it can be accepted for publication.
Author Response
-
Thank you for your valuable comments.
Reviewer 4 Report
The Authors need to clarify the following concerns
1- The study was built based on the Smith and Schlather model, but the smith model was not published, How the Authors are sure about the reliability of the result of this model which was not reviewed or published and how their result was validated. Do you have any publication used the smith model or any validation on this model? Please explain who you are sure about the accuracy of the result. The other concern is the climate data used in this study which mainly gathered from https://climexp.knmi.nl/start.cgi website. How the Author are sure about the accuracy about the data used in this link, any evidence. How the data was gathered and how it was filtered how the image was proceed and so one.
Comments:
2- The figure 2 and figure 3 show the average precipitation and the air temperature from 1970 to 2020, which they do not make sense. To make sense either you need another linear trend graph for each one to present the precipitation and the air temperature through the years from 1970 to 2020 or present at least two figures, one for 1970 and the other for 2022 to both factors. Then we can see the different and how the amount of these two factors incident to the case study.
3- Explain each symbol of the equations under each equation
4- Please use the average air temperature and the average precipitation throughout the manuscript.
5- Page 4 put citation number for (Usman and Reason, 2004)
6- In the text it was cited Ribatet in the reference was cited as Ribattet [11], make a consistence reference
7- Page 6 put number to Davison et al., 2012) reference
8- The f reference [20] not complete Ribatet, M. A user’s guide to the spatial extremes package, EPFL, Lausanne, Switzerland, Accessed on
9- Page 7 put a citation number for (Saunders et al. 2021),
- Smith and Schlather, were stated in the literature and in the abstract but the citation starts in the methodology, you have to cite them in the literature review.
Author Response

(The authors gave the same response as above.)

Reviewer 5 Report
This is a good project. The study examines the temperature's extreme dependence using gridded data. Incorporating rainfall data through two max-stable procedures. The researchers indicate that no other study has been conducted in South Africa's Lowveld region of the Limpopo province. The paper provides a modeling framework to help decision-makers in agriculture, hydrology, and other relevant sectors reduce the risk of drought brought on by extremely high temperatures and very little precipitation. Nonetheless, although a good, interesting, and important study is a bit confusing. It is difficult to follow and should be clearer. Figure 7-8 should be expanded since are difficult to read. Other than that, it is a good study that provides much information to scientist, the government, and farmers.Author Response
-
Thank you for your valuable comments, please check the attachment.

Round 2
Reviewer 1 Report
Thank you for your revised manuscript. The readers would appreciate it if you made these figures clear to read. The font size should be increased. Could you explain in the article why you always present a result of a study area inside a huge region?

Reviewer 4 Report
The Authors address the comments.